# SoLAR: Surrogate Label Aware GNN Rewiring

## Abstract

Rewiring the input graph of graph neural networks (GNNs) has been proposed as a pre-processing step to address issues like over-squashing and over-smoothing. However, most existing techniques rely solely on topology-based modifications, neglecting performance-critical node label information. To fill this gap, we propose SoLAR (Surrogate Label Aware Rewiring), a method that rewires the graph based on predicted node labels from a surrogate model. We prove its effectiveness in a theoretically tractable setting highlighting two key mechanisms that enable its success. The first is a denoising effect, while the second is a novel knowledge distillation-inspired process, where information from a surrogate model is encoded into the graph structure. Extensive experiments demonstrate consistent improvements of SoLAR across various datasets. Notably, the best surrogate models arise from iterative SoLAR, and reusing the same model class is a competitive strategy.

## 1 Introduction

Graph Neural Networks (GNNs) have emerged as powerful tools for analyzing and learning from graph-structured data, which is ubiquitous in real-world applications. Many complex systems can be naturally represented as graphs, and GNNs excel at capturing the relational information inherent in these data.We focus in this work on node classification, a fundamental task in graph learning which has numerous practical applications ranging from user profiling and interest prediction in social networks (Purificato et al., 2023), warning potentially infected nodes during a pandemic (Tomy et al., 2022; Burkholz & Quackenbush, 2021) to predicting protein functions within protein-protein interaction networks (Jha et al., 2022), to give a few examples.

Despite their success, GNNs face challenges like over-smoothing (Li et al., 2019; Oono & Suzuki, 2020) and over-squashing (Alon & Yahav, 2021; Topping et al., 2022), which limit their effectiveness on complex graph structures. Several studies argue that the input graph plays a crucial role during training, influencing the predictions even when the graph structure is uninformative (Bechler-Speicher et al., 2024). This suggests that graph rewiring as a pre-processing step to obtain a suitable computational structure holds great promise, and recent studies have proposed different rewiring criteria (Nguyen et al., 2023; Jamadandi et al., 2024) —mostly based on topological modifications such as the spectral gap, overlooking the importance of node label information.

But which edges should be rewired? According to Yang et al. (2024), GNN training dynamics tend to align with the structure of the graph, suggesting that an optimal input graph would closely align with the label distribution. In line with these findings, we study a theoretical setting in which GNN accuracy explicitly depends on the homophily level of the learning task and thus the tendency of connected nodes to share similar labels. The empirical strong correlation between homophily and message-passing GNN performance (Ma et al., 2022) has also motivated other efforts to improve homophily. As homophily cannot be measured without test labels, it is typically promoted during training (Jiang et al., 2024; Dai et al., 2022), or substituted by similarity measures (Bi et al., 2024).

In this work, we therefore ask the following questions: Can a better input graph be discovered such that it aligns more closely with the test labels without having access to them? Can this improve performance over the original graph, even when the modifications are dictated by the original predictions? We further study how a model can encode and transmit additional information by modifying its input graph, so that a second model can use it to outperform the first model. Moreover, a similar mechanism should be effective for both homophilic and heterophilic graphs.

To positively answer these questions, we propose SoLAR (Surrogate Label Aware Rewiring), a method that rewires the input graph to increase the predicted homophily based on a surrogate model —that is, by adding same-prediction edges and/or deleting different-label edges—, which is then used as the message-passing structure for a second model. We prove that rewiring based on predicted homophily can improve true homophily, and therefore accuracy. We show this to be true empirically and theoretically for both homophilic and heterophilic graphs. In addition, we demonstrate that the modified input graph can transfer extra information from the first model to the second one, ultimately leading to improved overall performance. This could be interpreted as a novel form of knowledge distillation, where information from a teacher model is encoded in the input graph of a student model. Yet, as our theory suggests, SoLAR rewiring goes beyond knowledge distillation, as the resulting model can learn more than the combination of the surrogate and the original model. We evaluate SoLAR on various GNN benchmarks with different model combinations, both in a one-shot approach and an iterative prediction-pruning process, and find performance boosts that align with our theory.

In summary, our **contributions** are as follows:

1. We propose SoLAR, a predicted-label aware rewiring mechanism that leverages a surrogate GNN model's predictions to modify the input graph of a second model. Iterative SoLAR, which alternates between model training and graph rewiring cycles, yields further performance boosts.

2. We develop a theoretical framework to show that rewiring based on *predicted homophily* will increase the true homophily and, consequently, improve GNN accuracy. Our analysis, grounded in mean field theory, finds this holds for both homophilic and heterophilic graphs.

3. SoLAR can be interpreted as a form of knowledge distillation, where information from the surrogate (teacher) model is encoded into the input graph to enable a subsequent (student) model to outperform the surrogate's performance. This mechanism offers a novel approach to information transfer between GNNs, namely through graph rewiring. Our theory highlights another mechanism that explains why, beyond knowledge distillation, the resulting GNN model can perform better than a combination of the original and the surrogate model.

4. We provide comprehensive experimental validation for SoLAR on a diverse set of benchmark datasets for both homophilic and heterophilic graphs. Our results show consistent improvements over existing baselines for graph rewiring.

## 1.1 RELATED WORK

**Graph rewiring.** Real-world graphs often contain noise or sub-optimal connections, leading to challenges such as over-squashing (Alon & Yahav, 2021; Topping et al., 2022; Giovanni et al., 2023), where bottlenecks cause an exponential amount of information to be squashed and potentially lost, and over-smoothing (Li et al., 2019; NT & Maehara, 2019; Oono & Suzuki, 2020; Zhou et al., 2021; Keriven, 2022), where nodes become more indistinguishable as the depth of the network increases. Graph rewiring is regarded as a standard strategy for addressing these challenges. Our work focuses on modifying the graph's edges prior to training, which can be done using a variety of criteria. For example, some methods aim to maximize the spectral gap to improve connectivity (Karhadkar et al., 2023) or to maximize other measures to mitigate over-squashing (Nguyen et al., 2023). While primarily edge additions have been considered, deletions also achieve competitive results, in particular on heterophilic graphs, where nodes tend to be connected to nodes with different labels (Jamadandi et al., 2024). Even though GNNs should learn to cut ties with neighbors when it aids their performance, in practice they often face difficulties in doing so (Mustafa et al., 2023; Mustafa & Burkholz, 2024), which explains why edge deletions can also help.

**Promoting homophily based on soft labels.** The optimization of homophily by means of soft label predictions has been empirically explored in the context of self-training (Li et al., 2018), where the training set is repeatedly enlarged based on the confidence of the pseudo-labels and the same model is trained. A variant of this approach (Nagarajan & Raghunathan, 2023) enhances the observed tendency of a graph, making homophilic graphs more homophilic and heterophilic graphs more heterophilic. Contrarily, our proposed approach follows a different principle: firstly, it is more general as it allows for different kinds of surrogate and student models and thus proposes a novel variant of knowledge distillation. Furthermore, the training data remains the same as we only use the predictions of the surrogate model to modify the edge structure. Our approach can also be repeated for a flexible number of iterations, and promotes homophily in both homophilic and heterophilic settings.

(1.1) Train Ⓐ  (1.2) Predict ●●  (2) SoLAR  (3.1) Train Ⓑ  (3.2) Predict ●●

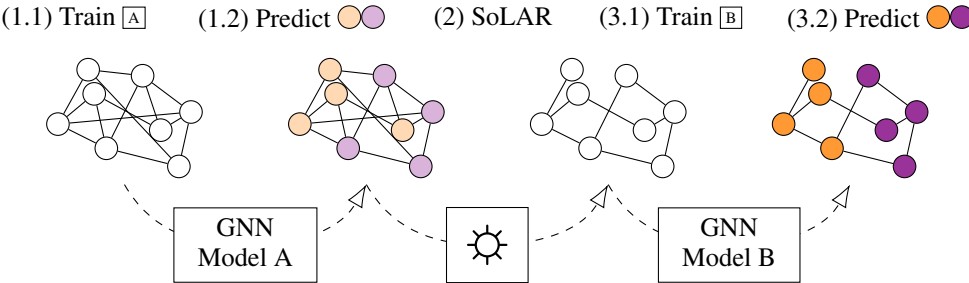

Figure 1: SoLAR: Surrogate Label Aware Rewiring. Model Ⓐ is trained (1.1) and used to predict its test labels (1.2). Its graph is then rewired (2) based on these predictions: adding same-class and/or deleting different-class edges. Model Ⓑ is trained on the new input graph (3.1). This can be used to test performance (3.2), but also to circle back to step (2) for Iterative SoLAR.

**Knowledge distillation.** Although we highlight that one of the key mechanisms for SoLAR's effectiveness is knowledge distillation, there are notable differences between our strategy and approaches related to knowledge distillation and pre-training. We do not have a separate objective function that tries to minimise the prediction discrepancy between the teacher and the student model(Yang et al., 2020; 2023; Tian et al., 2023) nor do we use context prediction or attribute masking (Hu et al., 2020) usually used in pre-training GNNs. SoLAR proposes a novel way to share knowledge, as the surrogate (teacher) model encodes its predictions in the input graph of another (student) model. Yet, the performance-boosting mechanism of SoLAR goes beyond the information transfer or knowledge distillation effect, as the resulting model can achieve a performance greater than the combination of the surrogate and the original model.

## 2 SoLAR: Surrogate Label Aware Rewiring

**Basic setup and notation.** In message-passing Graph Neural Networks (GNNs) (Gori et al., 2005; Scarselli et al., 2009), nodes exchange and aggregate information from their neighbors over multiple iterations, where each iteration corresponds to a graph neural network layer. This process enables GNNs to learn node representations that are used to solve tasks at the node or graph level. A typical setting for node classification is transductive or semi-supervised learning, where the input graph —and thus all nodes and edges— remains fixed throughout training and testing. More formally, let $\mathcal{G} = (\mathcal{V}, \mathcal{E})$ be an undirected and unweighted graph with $|\mathcal{V}|$ nodes and $|\mathcal{E}|$ edges. The adjacency matrix $A \in \mathbb{R}^{|\mathcal{V}| \times |\mathcal{V}|}$ encodes the graph topology. The goal in transductive node classification is to correctly classify the labels of the nodes in the test set by learning from nodes in the training set as well as their neighbors, whose labels are not necessarily known (but usually their features). That is, given a set of nodes $\mathcal{V}_{train}$ with available labels $\mathcal{Y}_{train}$, we need to predict the labels of nodes $\mathcal{V}_{test} = \mathcal{V} \backslash \mathcal{V}_{train}$. To solve the task, we train any message-passing GNN model that operates on node features $X$ and on the adjacency matrix $A$ of the input graph, or, more commonly, on the degree normalized adjacency matrix with added self-loops: $\hat{A} = \tilde{D}^{-1/2}(A + I)\tilde{D}^{-1/2}$, where $D$ denotes a diagonal matrix that carries the degrees $d_i$ of nodes $i \in \mathcal{V}$ and $I$ the identity matrix.

**GNN architectures.** In the message-passing paradigm, each layer of the network obtains node representations as a learnt function of the previous layer's representation and the node's aggregated neighbourhood. While several types of aggregation schemes are possible, we center our study on mean aggregation models, but SoLAR applies to general GNNs. Our main baseline is the Graph Convolutional Neural Network (GCN) (Kipf & Welling, 2017; Chen et al., 2021), where node representations in Layer $l$ take the following form:

$$H^{(l+1)} = f(H^{(l)}, A) = \sigma(\hat{A}H^{(l)}\Theta^{(1)}), \quad h_i^{(l+1)} = \sigma\left(\sum_{j \in N(i)} \frac{1}{\sqrt{|N(i)||N(j)|}}h_j^{(l)}\Theta^{(l)}\right),$$

where $\sigma(\cdot)$ is a non-linear activation function such as a ReLU and $\Theta$ the learnt weight matrix. This is equivalent to computing the normalized sum of each node $i$'s neighbourhood $N(i)$ (which includes $i$). The second model class that we consider is GATv2 (Brody et al., 2022), an improvement over

Graph Attention Networks (GATs) (Veličković et al., 2018). Its self-attention ($a$) applies learnable weights to neighbors, and is therefore considered more powerful than GCNs. Concretely, GATv2 takes the following form:

$$h_i^{(l+1)} = \sigma \left( \sum_{j \in N(i)} \alpha_{ij}^{(l)} \Theta^{(l)} h_j^{(l)} \right), \ \alpha_{ij}^{(l)} = \frac{\exp(e_{ij}^{(l)})}{\sum\limits_{k \in N(i)} \exp(e_{ik}^{(l)})}, \ e_{ik}^{(l)} = a^\top \sigma \left( \Theta^{(l)} (h_i^{(l)} || h_j^{(l)}) \right)$$

**SoLAR.** To increase the homophily of a learning task, we propose SoLAR, which uses predictions made by a surrogate GNN model as proxy labels for rewiring the input graph of a second model, as illustrated in Figure 1. Specifically, the process works in three stages. In the first stage, we instantiate the surrogate GNN model $f_{surrogate}(\mathcal{G}, \mathbf{\Theta})$ and train it to convergence obtaining a set of predicted labels. In the second stage, we use the predicted labels, $\mathcal{Y}_{surrogate}$, to rewire the graph by either deleting (predicted) inter-class edges and/or adding (predicted) intra-class edges to obtain a rewired graph $\hat{\mathcal{G}} = (\mathcal{V}, \hat{\mathcal{E}})$. We use the predictions only on the test and validation sets, as we already have access to the ground truth labels on the train set. In the last stage, we instantiate a second 'training' or 'student' GNN model $f_{train}(\hat{\mathcal{G}}, \mathbf{\Theta})$, which operates on the rewired graph. The above outlined rewiring can either be applied in a one-shot way, or iteratively, where the model from the previous round becomes the surrogate model of the next round.

## 3 CONCEPTUAL ANALYSIS

At first glance, it is not apparent why SoLAR should improve the performance of a model beyond a knowledge distillation mechanism, where a surrogate model has access to information that complements another model. If we rewired the input graph based on a surrogate model $\mathcal{M}$ and then retrained the same model $\mathcal{M}$ on the rewired graph, would we not simply enhance our original findings and, for instance, increase our certainty but not gain additional information? Against this intuition, our conceptual analysis of the next sections negates this question and highlights a simple mechanism for how SoLAR can still obtain performance gains in this setting. In fact, we show that $\mathcal{M}$ does not even need to be retrained on the rewired graph. Only deleting edges according to SoLAR can improve the node classification task.

**Theoretical setup.** We study a simplified, theoretically tractable 2-class classification problem, where $pn$ of the $n$ nodes are assigned to class $c = c_1$ and $(1-p)n$ to class $c = c_2$. For simplicity, we consider 1-dimensional independently, normally distributed features as visualized in Figure 2(b), i.e., $X_i^{(0)} \sim \mathcal{N}(0, 1)$ given that a node $i$ has class $c_i = c_1$ and $X_i^{(0)} \sim \mathcal{N}(\mu, 1)$ if it has class $c_i = c_2$. Their exact distribution and dimensionality are not relevant to our general argument. We only require their distributions to partially overlap to create a sufficiently difficult learning problem, where nodes get misclassified (as illustrated by the shaded regions in the figure). In addition, we assume that nodes are connected by a graph with normalized adjacency matrix $\tilde{A} = (D + I)^{-1}(I + A)$.

### 3.1 THE CONCEPTUAL BASIS BEFORE SoLAR REWIRING

**Classification without graph.** In this context, without considering any graph structure, it is easy to verify that the Bayes optimal decision threshold to classify nodes based on their features, which maximizes the expected classification accuracy, is $\theta = \mu/2$. Accordingly, nodes with features $X_i^{(0)} \leq \theta$ receive the predicted label $\hat{c}_i = c_1$ and $c_2$ otherwise. The resulting accuracy is binomially distributed as $\mathcal{A} \sim \frac{1}{n} \text{Bin}(n, \Phi(\frac{\mu}{2}))$ with expected value $\mathbb{E}(\mathcal{A}) = \Phi(\frac{\mu}{2})$, where $\Phi$ denotes the cumulative distribution function of the standard normal. Figure 3 labels this scenario as 'initial'. Can this approach be improved with the help of a known graph structure?

**Classification with mean aggregation.** As an analytically tractable proxy of a GNN layer (with mean aggregation), we consider one round of mean aggregation where we do not learn the aggregation weights. Specifically, we consider node features that are updated as $X_i^{(1)} = 1/(d_i + 1)(X_i^{(0)} + \sum_{j \in N(i)} X_j^{(0)})$. The question is whether these updated features are better suitable for solving the node classification problem, which is investigated by the following theorem.

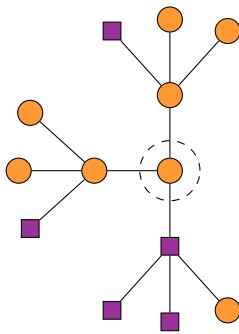

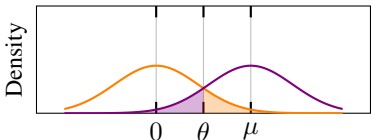

(b) Distribution of features before aggregation for class $c_1$ (orange, $\sim\mathcal{N}(0,1)$) and class $c_2$ (purple, $\sim\mathcal{N}(\mu,1)$).

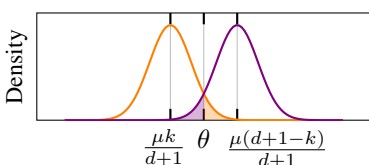

(a) Model for $d = 3$, $k = 1$, where orange circles are $c_1$ nodes and purple squares are $c_2$ nodes. The central node is highlighted.

(c) Distribution of features after aggregation for class $c_1$ (orange, $\sim\mathcal{N}\big(\frac{\mu k}{d+1}, \frac{1}{d+1}\big)$) and class $c_2$ (purple, $\sim\mathcal{N}\big(\frac{\mu(d+1-k)}{d+1}, \frac{1}{d+1}\big)$).

Figure 2: For each $(d, k, \mu)$ we instantiate 15000 graphs like the one in 2(a) with initial features drawn from the distributions in 2(b). After one step of aggregation, they follow the ones in 2(c).

**Theorem 1.** *After one step of mean aggregation, the expected accuracy of node classification is*

$$\mathbb{E}(\mathcal{A}) = \frac{1}{n} \sum_{i, c_i = c_1} \Phi\left(\frac{\theta d_i^+ - k_i \mu}{\sqrt{d_i^+}}\right) + \frac{1}{n} \sum_{i, c_i = c_2} \left(1 - \Phi\left(\frac{\theta d_i^+ - (d_i^+ - k_i)\mu}{\sqrt{d_i^+}}\right)\right),$$

*where $\theta$ is the decision threshold, $k_i$ the number of neighbors that have the opposite class of $i$, and $d_i^+ = d_i + 1$ denotes the degree of a node including self-loops.*

*Proof.* As the initial features are normally distributed, their sum follows a joint multivariate normal distribution, which is simple to derive with probabilistic calculus. The vector $X^{(1)}$ follows the multivariate normal distribution $\mathcal{N}(\mu^{(1)}, \Sigma^{(1)})$ with mean $\mu^{(1)} = \tilde{A}\mu^{(0)}$, where $\mu^{(0)}$ denotes the vector of initial means, i.e., its first $p \cdot n$ entries are zero and the last $(1 - p) \cdot n$ entries are $\mu$. The covariance is given by $\Sigma^{(1)} = \tilde{A}\tilde{A}^\top$. Accordingly, the marginal feature distribution of a node of class $c_1$ is $X_i^{(1)} \sim \mathcal{N}((\mu k_i)/d_i^+, 1/d_i^+)$, while it is $X_i^{(1)} \sim \mathcal{N}((\mu(d_i^+ - k_i))/d_i^+, 1/d_i^+)$ for a node of class $c_2$. The stated formula follows from the fact that the expected accuracy is simply the average of all probabilities that a node is correctly classified. $\qquad\square$

The above derivation provides clear insights into the potential benefits of neighborhood aggregation, as the variance of the features is reduced from 1 to $1/d_i^+$, so that the features get more concentrated around their mean, which makes it potentially easier to differentiate the classes.

**Benefits of homophily.** Unfortunately, the feature means get potentially diluted, as the means of nodes of different classes move closer together: they are shifted by $k_i/d_i^+$, which is determined by the overall homophily of the task. $k_i$ is generally small if nodes are primarily connected to nodes with the same label; thus, the mean shift of the distributions is negligible. As a remark, note that, in this simple 2-class scenario, also extreme heterophily —where nodes are almost exclusively connected to nodes of a different class— would be helpful. The attributed label decision would simply need to be reversed, but nodes would receive well separable features. Figure 2(c) visualizes our insight for a highly symmetric and homophilic scenario (with $p = 0.5$ and $k_i = k$), where the Bayes optimal threshold remains $\theta = \mu/2$. Note that the misclassification rate (i.e. the size of the shaded area) is smaller compared with no mean aggregation in Figure 2(b). In the following sections, we label the discussed scenario of one-step mean aggregation based on the original graph as 'one-step'.

**SoLAR rewiring increases dependencies.** A general analysis of the effect of SoLAR rewiring is challenging, as it introduces and requires capturing higher-order dependencies of feature distributions. After one step of mean aggregation, the node features are already not distributed independently anymore, and neither are the resulting node labels. To obtain a theoretically tractable setting, instead

of considering arbitrary finite graphs $A$, we further focus our analysis on the (heterogenous) mean field limit, where we send the number of nodes to infinity and can study the effect of SoLAR pruning.

## 3.2 MEAN FIELD ANALYSIS

The mean field limit, also known as branching process approximation, is a common tool in complex network science and theoretical physics to obtain theoretically tractable insights (Dorogovtsev et al., 2008; Gleeson & Cahalane, 2007; Burkholz & Schweitzer, 2018) and emerges in the large graph limit of the configuration model (Molloy & Reed, 1995; Newman et al., 2001), where the graph structure is characterized by a degree distribution $p_D$ (and, potentially, degree-degree correlations of connected nodes, which we do not need here). The graph neural tangent kernel (Du et al., 2019) can be considered as a special case of similar assumptions. Most importantly, in the limit $n \to \infty$, the graph structure becomes locally tree-like, which allows us to treat the states of neighbors as independent, and often also results in a good approximation of sparse, finite graphs in practice.

**Theorem 2.** *In the heterogenous mean field limit, where nodes are equipped with degree distribution $p_D$, and $k$ out of the $d$ neighbors have a different label with probability $p(d, k)$, the expected accuracy after one round of mean aggregation is*

$$\mathbb{E}(\mathcal{A}) = \sum_d p_D(d) \sum_{k=0}^{d} p(d, k) \left[ p\Phi\left( \frac{\theta d^+ - k\mu}{\sqrt{d^+}} \right) + (1 - p)\left( 1 - \Phi\left( \frac{\theta d^+ - (d^+ - k)\mu}{\sqrt{d^+}} \right) \right) \right].$$

The derivation of this formula is presented in the appendix (§A), but is straightforward as it follows similar arguments as the proof of Theorem 1 and averages over the relevant cases. Similarly to our previous analysis, we see that nodes with high degree and few neighbors of a different class have the highest chance of getting classified correctly.

To obtain the Bayes optimal threshold $\theta$ for this setting, we would need to differentiate the above accuracy with respect to $\theta$, which would require us to solve a fixed point equation, whose result is difficult to interpret. To further simplify our analysis (for improved interpretability but general insights), we further specialize our study to regular random graphs. An exemplary two-step neighbourhood of this setting is illustrated in Figure 2(a). Because of the symmetry in this graph, the expected accuracy for this representative central node after a step of aggregation also corresponds to the overall average accuracy, as the following theorem states.

**Theorem 3.** *In the homogeneous mean field case, where each node has the same degree $d$ and is connected to a fixed number $k$ of neighbors with a different class label and the class memberships is balanced with $p = 0.5$, the expected accuracy after one round of mean aggregation is*

$$\mathbb{E}(\mathcal{A}) = \Phi\left( \frac{\theta d^+ - k\mu}{\sqrt{d^+}} \right). \tag{1}$$

*The Bayes-optimal classification threshold is $\theta = \frac{\mu}{2}$.*

*Proof.* Setting the degree distribution in by $p_D(d) = 1$ and $p_D(x) = 0$ otherwise, setting $p(d, k) = 1$ and $p(x, y) = 0$ otherwise, and using $p = 0.5$ in Theorem 2 leads to the stated expression, as the classification accuracy of nodes of different classes is symmetric $\Phi\left( \frac{\theta d^+ - k\mu}{\sqrt{d^+}} \right) = 1 - \Phi\left( \frac{\theta d^+ - (d^+ - k)\mu}{\sqrt{d^+}} \right)$. From the symmetry, it also follows that the Bayes optimal threshold is $\theta = (\mu_1 + \mu_2)/2$, where $\mu_1 = k\mu/d^+$ and $\mu_2 = (d^+ - k)\mu/d^+$. Setting the derivative of the expected accuracy above to zero would lead to the same conclusion. $\square$

We thus have obtained a setting, in which we can analyze the effect of deleting edges according to the SoLAR criterion. Thus, all edges between nodes that have different predicted labels after one round of mean aggregation are deleted. Mean aggregation with respect to the new input graph then defines our SoLAR accuracy.

**Theorem 4.** *After one round of mean aggregation and deleting edges between nodes that do not share the same predicted label, the expected SoLAR accuracy of mean aggregation with respect to the rewired graph becomes*

$$\mathbb{E}(\mathcal{A}) = \mathbb{P}\left( X_0^{(0)} + \sum_{i=1}^{d} S_i X_i^{(0)} \leq \left( \sum_{i=1}^{d} S_i + 1 \right) \frac{\mu}{2} \right),$$

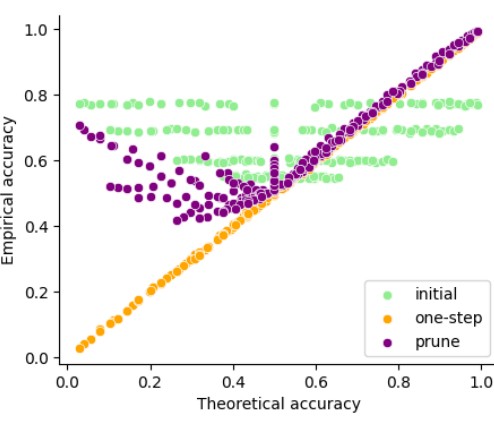

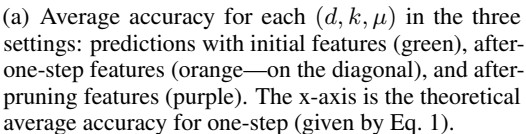

(a) Average accuracy for each $(d, k, \mu)$ in the three settings: predictions with initial features (green), after-one-step features (orange—on the diagonal), and after-pruning features (purple). The x-axis is the theoretical average accuracy for one-step (given by Eq. 1).

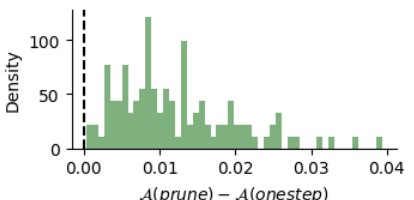

(b) Homophilic $(d, k, \mu)$ combinations.

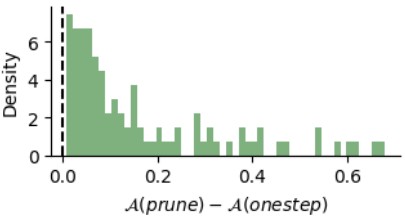

(c) Heterophilic $(d, k, \mu)$ combinations.

Figure 3: Homogenous mean field simulations for the frequency of correct central node prediction after SoLAR pruning. In 3(a), each point shows the average accuracy of each $(d, k, \mu)$ combination in comparison to one step of aggregation. The histograms of the difference between 'prune' and 'one-step' are plotted in Figures 3(b) (homophilic) and 3(c) (heterophilic).

*where $S_i \in \{0, 1\}$ indicates whether the respective node is pruned ($S_i = 0$) or not ($S_i = 1$). The messages $Z_i = S_i X_i^{(0)}$ sent by neighbors are only independent given the initial node feature $X_0^{(0)}$ and neighbor features $X_i^{(0)}$. We have $S_i = 1$ if $X_0^{(0)} + \sum_{i=1}^{d} X_i^{(0)} \leq \frac{\mu}{2} d^+$ and $X_i^{(1)} = (X_0^{(0)} + X_i^{(0)} + Y_i) \leq d^+ \mu/2$ or if $X_0^{(0)} + \sum_{i=1}^{d} X_i^{(0)} > \frac{\mu}{2} d^+$ and $X_i^{(1)} = (X_0^{(0)} + X_i^{(0)} + Y_i) > d^+ \mu/2$, where $Y_i \sim 0.5 \mathcal{N}(\mu k, d-1) + 0.5 \mathcal{N}(\mu(d-k+1), d-1)$. $Z_i = S_i = 0$ otherwise.*

The proof is presented in the appendix (§A). As the expression does not have a simple closed form solution, we evaluate it approximately by sampling and report the results in Figure 3.

**SoLAR in homogeneous setting.** Note that Theorem 4 encompasses different cases in which the expected label accuracy of a node is increased or decreased. The question is which mechanisms are dominating: it depends on the specific choices of $(d, k, \mu)$, which we vary in our evaluations. For each combination of $d \in \{1, \ldots, 9\}$, $k \in \{0, \ldots, d\}$, and $\mu \in \{0.25, 0.5, 1, 1.5\}$, we draw 15000 instances of a tree-like graph, drawing the node's features from the distributions respecting their random classes, $\mathcal{N}(0, 1)$ and $\mathcal{N}(\mu, 1)$. Figure 3(a) reports the average accuracy with respect to the theoretical accuracy of the original model (Eq. (1)) of each $(d, k, \mu)$ combination in the three relevant settings: initialization —which shows how informative the original features are—, after one step of aggregation —which follows Eq. (1)—, and after SoLAR pruning. We observe that heterophilic cases (on the left of $x = 0.5$) behave differently than homophilic ones (on the right of $x = 0.5$). In heterophilic cases, the original features (green) are more informative than the aggregated features (orange). Thus, pruning some of the neighbours assigns more importance to the self-feature. In homophilic cases, the neighbourhood usually agrees with the self-feature and the class, but pruning will aid in special cases where the neighbourhood noise is detrimental. This overall improves the average accuracy for all $(d, k, \mu)$ combinations, as illustrated in Figures 3(b) and 3(c). For instance, the probability is higher to have a same-class neighbour with a feature that resembles one from the other class than to have a self-feature with this property. Therefore, there are more cases in which a conflicting neighbour is detrimental to the aggregation and its removal aids the classification task.

### 3.3 REAL-WORLD GRAPH SIMULATIONS OF SOLAR

To account for degree-heterogeneous real-world graph structures, we perform a similar set of experiments on two homophilic (Cora (McCallum et al., 2000), Citeseer (Sen et al., 2008)) and two

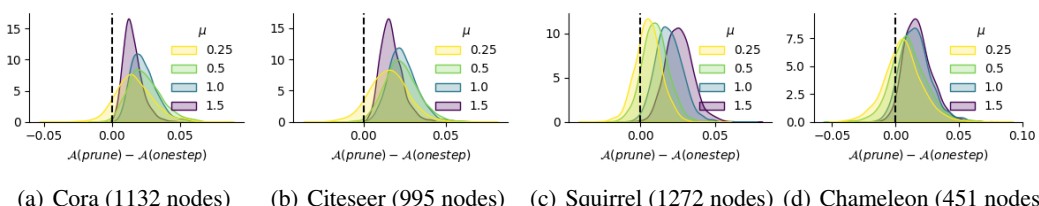

(a) Cora (1132 nodes)    (b) Citeseer (995 nodes)    (c) Squirrel (1272 nodes)    (d) Chameleon (451 nodes)

Figure 4: Distribution of accuracy differences between the after-pruning and after-one-step settings for different $\mu$ for 1500 seeds each. Positive values indicate that pruning was more successful.

heterophilic (Squirrel, Chameleon (Platonov et al., 2023b)) graphs and focus on the subgraph that is induced by the two largest classes. This preserves their homophilic tendency. The nodes' initial features are instantiated according to their class and are thus drawn from $\mathcal{N}(0,1)$ or $\mathcal{N}(\mu,1)$, as before. We generate 1500 of these instances per graph and choice of $\mu \in \{0.25, 0.5, 1, 1.5\}$. Then, we count the number of correctly predicted nodes using the decision threshold $\frac{\mu}{2}$ in the three settings described before. The number of nodes for each graph is 1132 (Cora), 995 (Citeseer), 1272 (Squirrel), and 451 (Chameleon).

Figure 4 shows the difference between the accuracy resulting from SoLAR pruning and the accuracy after one-step mean aggregation. The mostly positive values indicate that SoLAR usually improves accuracy. In homophilic graphs, $\mu$ changes the variance of the distributions. In the heterophilic cases, it affects more evidently the mean of the distributions. In both cases, SoLAR pruning is statistically more successful than neighborhood aggregation based on the original graph.

Next, we examine what kind of edges are most usually recovered or corrupted comparing the one-step aggregation with the SoLAR predictions. We expect that pruning an edge improves more than it corrupts; meaning that the nodes it connects have their predictions corrected if this edge disappears. Note that we count the same node multiple times depending on its degree, which is not accounted for in the following analysis but can influence the results of Figure 4.

We first consider the homophilic graph Cora. Taking a mean across all considered $\mu$ values, we find that on average 21% of the edges are pruned, of which 92% one node was originally correctly predicted and one not. This means we have pruned same-label edges, but some should have been confusing for one of the neighbours. In 22% of cases, we improve the wrong node's prediction after pruning. In 68% of the cases, we maintain the original predictions, in 5% we flip both labels, and in 4% we corrupt the originally right node. This indicates where the performance improvements come from: we are denoising the neighbourhoods of some of the nodes (22% > 4%). Among the non-pruned edges (79% of all edges), SoLAR leaves the corresponding labels of 97% unchanged.

Let us now consider the heterophilic case based on Squirrel, where we prune 20% of edges on average. Of them, 24% had both nodes rightly predicted with only one-step neighborhood aggregation, 26% had both wrongly predicted, and the remaining 50% had one well predicted and one wrongly predicted. It is clear that the distribution of edge predictions is different from the previous homophilic case because of the heterophily. When both nodes adjacent to an edge were rightfully predicted, pruning corrupts one of them in 16% of cases, and both in 6%. When both have been wrongly predicted, pruning improves one in 28% of the cases, and both in 12%. This leaves us with more improved than corrupted cases after SoLAR pruning. As for the edges with one good and one bad prediction (the other 50%), we improve the wrong one in 12% and corrupt the right one in 10% of the cases, which is comparable. The trends are similarly maintained in Chameleon, but this last comparison yields 14% against 7%, which is again favorable for pruning. More detailed results that distinguish the separate values of $\mu$ and the 4 studied datasets can be found in §B.

## 4 EXPERIMENTS

We perform node classification on the following homophilic datasets: Cora (McCallum et al., 2000), Citeseer (Sen et al., 2008) and Pubmed (Namata et al., 2012), Co-author CS, Physics and Amazon Photos (Shchur et al., 2019) and consider the heterophilic graphs Chameleon, Squirrel, Actor and the

Table 1: Node classification using one-shot SoLAR on large heterophilic graphs.

| Method | Roman-Empire | Amazon-Ratings | Penn94 |
|---|---|---|---|
| GCN | 77.74±0.60 | 47.66±0.54 | 82.29±0.77 |
| GATv2 | 82.52±0.50 | 47.66±0.95 | 81.85±3.02 |
| GCN+FoSR | 73.60±1.11 | 49.68±0.73 | 69.73±7.83 |
| GATv2+FoSR | 81.88±1.07 | 51.36±0.62 | 72.56±5.55 |
| GCN☼GCN+Delete | 80.90±0.14 | 50.30±0.09 | 83.59±1.40 |
| GCN☼GCN+Add | 81.13±0.21 | 49.86±0.11 | **83.65±1.69** |
| GATv2☼GATv2+Delete | **84.32±0.80** | **52.06±0.00** | 83.58±1.60 |
| GATv2☼GATv2+Add | **84.27±0.40** | **52.08±0.09** | 83.60±1.32 |

Table 2: Node classification on homophilic graphs using one-shot SoLAR.

| Method | Cora | Citeseer | Pubmed | CS | Physics | Photo |
|---|---|---|---|---|---|---|
| GCN | 87.94±3.35 | 79.38±3.48 | 81.99±1.42 | 92.44±0.67 | 93.64±0.16 | 92.89±1.23 |
| GATv2 | 89.13±3.13 | 81.92±4.81 | 81.83±1.04 | 91.90±1.59 | 94.07±0.44 | 91.22±2.18 |
| GCN+FoSR | 88.74±2.70 | 79.48±3.77 | 82.22±1.24 | 93.54±0.80 | 94.72±0.21 | 90.57±3.82 |
| GATv2+FoSR | 89.72±2.91 | 81.75±4.86 | 81.29±2.31 | 92.35±1.21 | 93.96±0.40 | 90.48±2.57 |
| GCN☼GCN+Delete | 90.17±2.82 | 82.22±4.01 | 82.61±1.16 | 93.00±0.21 | 93.96±0.10 | 93.75±0.99 |
| GCN☼GCN+Add | 90.06±2.56 | 83.26±4.44 | 83.05±2.50 | 92.46±0.56 | 95.47±0.31 | 92.13±0.32 |
| GATv2☼GATv2+Delete | 90.06±3.31 | 83.01±4.32 | 82.41±2.46 | **94.16±1.79** | **95.01±0.54** | 93.78±1.30 |
| GATv2☼GATv2+Add | 89.63±3.16 | 81.78±4.44 | 81.32±1.66 | 92.79±1.58 | 94.25±0.46 | 93.36±1.93 |
| GCN☼GATv2+Delete | 90.23±0.59 | 81.48±0.77 | 83.15±0.26 | 93.96±0.15 | 87.01±2.09 | **94.80±0.03** |
| GCN☼GATv2+Add | 90.01±0.58 | 81.42±0.85 | 82.29±0.31 | 93.41±0.22 | 84.61±2.51 | 94.30±0.05 |
| GATv2☼GCN+Delete | **90.42±0.65** | **83.93±0.90** | **83.20±0.28** | 93.38±0.26 | 92.08±0.62 | 94.56±0.04 |
| GATv2☼GCN+Add | **90.47±0.60** | 83.44±0.86 | 82.71±0.27 | 93.65±0.17 | 92.47±0.44 | 94.67±0.03 |

WebKB datasets consisting of Cornell, Wisconsin and Texas (Platonov et al., 2023b). Additionally, we study three large heterophilic graphs: Roman-empire and Amazon-ratings introduced in (Platonov et al., 2023b), and Penn94 (Lim et al., 2021). In contrast to the other rewiring approaches (Topping et al., 2022; Karhadkar et al., 2023; Nguyen et al., 2023; Giraldo et al., 2023; Jamadandi et al., 2024) that tune the number of rewired edges, we rewire all possible edges, which we can assess based on train set ground truth labels and predicted test and validation set labels.

We adopt 60/20/20 splits for training, validation and testing respectively. The final test accuracy is reported as an average over 100 splits of the data. (see §E for details and hyperparameters). The top performance is highlighted in bold. We compare GCN and GATv2 as baselines and in combination with FoSR (Karhadkar et al., 2023), PROXYADDMAX, PROXYDELMAX (Jamadandi et al., 2024), which add or delete edges that maximize a proxy of the spectral gap. Our proposed SoLAR rewiring method is denoted as $\mathcal{M}_1$☼$\mathcal{M}_2$, where the first model $\mathcal{M}_1$ provides the surrogate labels for rewiring and the second model uses the rewired graph for training on the downstream task. We report results for both predicted-inter-class edge deletions and predicted-intra-class edge additions. When deleting inter-class edges, we ensure we do not disconnect the graph and leave a few edges to preserve the original structural integrity of the graph.

**One-shot experiments.** Table 2 presents our results for homophilic graphs, and Tables 1 and 3 for heterophilic graphs. Evidently, our proposed rewiring boosts the GNN performance across all studied datasets. On large heterophilic datasets like Roman-empire and Amazon-ratings, especially GATv2☼GATv2 performs well. This supports our theoretical insight that SoLAR can improve performance even if the surrogate and student models belong to the same model class, which suggests benefits beyond pure knowledge distillation. In some homophilic cases, the combination of GATv2☼GCN is even stronger. These results indicate that a powerful model (like GATv2) is particularly effective in the position of the surrogate model, which has to be responsible for reliable rewiring decisions.

**Iterative SoLAR.** Multiple SoLAR cycles boost the performance even further, as shown by Table 4. It achieves the overall best result on 10 out of 12 datasets. Additional experimental results are reported in §D, where also more GNN architectures are considered.

Table 3: Node classification on heterophilic graphs using one-shot SoLAR.

| Method | Cornell | Texas | Wisconsin | Chameleon | Squirrel | Actor |
|---|---|---|---|---|---|---|
| GCN | 68.31±8.13 | 73.47±10.13 | 66.14±9.23 | 54.64±6.94 | 43.25±6.32 | 28.26±3.22 |
| GATv2 | 86.84±9.78 | 89.01±10.43 | 87.56±9.20 | 61.79±10.20 | 45.71±5.12 | 29.41±2.98 |
| GCN+FoSR | 71.64±9.80 | 73.93±10.23 | 65.85±7.73 | 54.40±6.58 | 42.80±6.40 | 28.66±3.21 |
| GATv2+FoSR | 76.12±6.51 | 78.15±7.81 | 74.08±9.01 | 46.48 ± 4.97 | 47.40±7.17 | 27.45±3.61 |
| GCN+ProxyAddMax | 67.57±1.71 | 81.08±1.75 | 70.00±1.61 | 56.74±0.90 | 33.26±0.39 | 27.57±0.22 |
| GCN+ProxyDelMax | 62.16±1.83 | 72.97±1.70 | 76.00±1.56 | 56.74±0.95 | 32.58±0.43 | 27.96±0.21 |
| GCN☼GCN+Delete | 68.35±8.54 | 74.12±9.89 | 67.85±7.14 | 57.19 ± 6.45 | 44.50±6.29 | 29.25±3.50 |
| GCN☼GCN+Add | 69.42±8.93 | 74.20±10.26 | 68.51±7.20 | 56.43 ± 6.16 | 44.04±6.34 | 28.16±3.22 |
| GATv2☼GATv2+Delete | **87.40±9.89** | **90.14±10.64** | **88.32±9.08** | **68.89±11.50** | **49.10±5.59** | 30.31±4.29 |
| GATv2☼GATv2+Add | 87.12±9.59 | 87.97±10.95 | 87.76±9.57 | 66.35±11.18 | 46.44±6.00 | 29.46±4.67 |
| GCN☼GATv2+Delete | 84.03±2.12 | 86.91±2.23 | 84.53±1.95 | 60.11±1.59 | 47.98±1.17 | 30.02±0.73 |
| GCN☼GATv2+Add | 83.11±1.88 | 85.01±2.10 | 85.96±1.70 | 54.99±1.42 | 43.17±1.10 | 30.09±0.93 |
| GATv2 ☼GCN+Delete | 78.63±2.01 | 84.65±2.12 | 77.65±1.86 | 68.60±2.20 | 47.89±1.36 | **30.91±0.94** |
| GATv2 ☼GCN+Add | 85.37±2.22 | 87.43±2.28 | 83.00±1.96 | 68.27±2.34 | 47.70±1.23 | 29.15±0.85 |

Table 4: Accuracy of iterative SoLAR for homophilic and heterophilic graphs

| Homophilic | Cora | Citeseer | Pubmed | CS | Physics | Photo |
|---|---|---|---|---|---|---|
| GCN | 87.94±3.35 | 79.38±3.48 | 81.99±1.42 | 92.44±0.67 | 93.64±0.16 | 92.89±1.23 |
| GATv2 | 89.13±3.13 | 81.92±4.81 | 81.83±1.04 | 91.90±1.59 | 94.07±0.44 | 91.22±2.18 |
| Best one-shot (Table 2) | 90.47±0.60 | 83.93±0.90 | **83.20±0.28** | 94.16±1.79 | **95.01±0.54** | **94.80±0.03** |
| GCN☼GCN+Delete | 90.72±0.56 | 82.61±0.88 | 82.86±0.42 | 94.23±0.20 | 94.48±0.18 | 94.23±0.24 |
| GCN☼GCN+Add | 91.95±0.59 | 82.29±0.88 | 82.98±0.41 | 94.41±0.22 | 94.51±0.18 | 94.24±0.27 |
| GCN☼GATv2+Delete | 92.20±0.68 | 84.49±0.91 | 83.06±0.38 | **94.85±0.19** | 94.90±0.16 | 93.59±1.09 |
| GCN☼GATv2+Add | **92.87±0.67** | **85.41±0.97** | 82.88±0.45 | **94.89±0.35** | 94.91±0.21 | 93.86 ± 1.38 |
| **Heterophilic** | **Cornell** | **Texas** | **Wisconsin** | **Chameleon** | **Squirrel** | **Actor** |
| GCN | 68.31±8.13 | 73.47±10.13 | 66.14±9.23 | 54.64±6.94 | 43.25±6.32 | 28.26±3.22 |
| GATv2 | 86.84±9.78 | 89.01±10.43 | 87.56±9.20 | 61.79±10.20 | 45.71±5.12 | 29.41±2.98 |
| Best one-shot (Table 3) | 87.40±9.89 | 90.14±10.64 | 88.32±9.08 | 68.89±11.50 | 49.10±5.59 | 30.91±0.94 |
| GCN☼GCN+Delete | 77.27±2.13 | 82.66±2.06 | 75.50±1.83 | 61.45±1.43 | 50.19±1.44 | 32.78±0.83 |
| GCN☼GCN+Add | 78.52±2.25 | 84.55±2.26 | 75.01±1.94 | 62.14±1.54 | 49.87±1.42 | 31.69±0.80 |
| GCN☼GATv2+Delete | **92.90±1.93** | **94.07±1.82** | **93.87±1.74** | **71.72±2.38** | **54.37±1.54** | 34.01±0.93 |
| GCN☼GATv2+Add | 81.00±2.15 | 87.41±2.12 | 81.54±1.87 | 65.23±1.94 | 49.14±1.36 | **34.61±1.18** |

## 5 DISCUSSION

Both our theoretical analysis (§3) and extensive experiments (§4) have established that our proposed graph rewiring strategy significantly boosts GNNs. SoLAR not only distills knowledge but also obtains models that can outperform the combination of the surrogate and initial model. It works seamlessly for both homophilic and heterophilic settings, contrary to methods which use non-robust feature similarity measures (Huang et al., 2020; Bi et al., 2024) or require expensive k-hop rewiring during training (Gutteridge et al., 2023). Different from methods that rely purely on topological characteristics, our approach optimizes homophily (cf. §C), a critical predictor of GNN performance. However, the impact of rewiring largely depends on the quality of the surrogate. If the predicted labels are too noisy, they might amplify issues that were already present in the initial model. In such scenarios, it could be interesting to take features and the uncertainty of predictions into account. Furthermore, SoLAR rewiring does not explicitly address over-squashing and over-smoothing, which relate to problems with trainability and information propagation. Combining SoLAR with topological considerations could be another direction for future extensions.

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

# APPENDIX

## A  PROOFS

**Theorem** (Theorem 2 in main paper). *In the heterogenous mean field limit, where nodes are equipped with degree distribution $p_D$, and $k$ out of the $d$ neighbors have a different label with probability $p(d, k)$, the expected accuracy after one round of mean aggregation is*

$$\mathbb{E}(\mathcal{A}) = \sum_d p_D(d) \sum_{k=0}^{d} p(d, k) \left[ p\Phi\left( \frac{\theta d^+ - k\mu}{\sqrt{d^+}} \right) + (1 - p) \left( 1 - \Phi\left( \frac{\theta d^+ - (d^+ - k)\mu}{\sqrt{d^+}} \right) \right) \right].$$

*Proof.* In the mean field limit, the expected accuracy can also be interpreted as the probability that a random node in the network is correctly classified. We call this random node also the center or focal node, as it is highlighted in Figure 2(a). Its neighbors are not connected and can be considered independent given the focal node. The probability that a focal node is correctly classified depends then on its class membership $C$ and its features $X^{(1)}$ after one round of mean neighborhood aggregation, which in turn depends on the focal node's degree $D$, and the number of different class neighbors

$K$. Thus, using the tower property of conditional expectation, these different variables have to be considered resulting in

$$\mathbb{E}(\mathcal{A}) = \mathbb{E}(\mathcal{A} \mid C = c_1)\mathbb{P}(C = c_1) + \mathbb{E}(\mathcal{A} \mid C = c_2)\mathbb{P}(C = c_2)$$

$$= \mathbb{E}(\mathcal{A} \mid C = c_1)p + \mathbb{E}(\mathcal{A} \mid C = c_2)(1-p) = \mathbb{P}(X^{(1)} \leq \theta \mid C = c_1)p + \mathbb{P}(X^{(1)} > \theta \mid C = c_2)(1-p)$$

$$= \sum_d \mathbb{P}(D = d) \left( \mathbb{P}(X^{(1)} \leq \theta \mid C = c_1, D = d)p + \mathbb{P}(X^{(1)} > \theta \mid C = c_2, D = d)(1-p) \right)$$

$$= \sum_d p_D(d) \sum_{k=0}^{d} p(d,k) \Big( \mathbb{P}(X^{(1)} \leq \theta \mid C = c_1, D = d, K = k)p$$

$$+ \mathbb{P}(X^{(1)} > \theta \mid C = c_2, D = d, K = k)(1-p) \Big).$$

Now, $\mathbb{P}(X^{(1)} \leq \theta \mid C = c_1, D = d, K = k)$ is a probability that we have derived before. Note that

$$X^{(1)} \mid D = d, K = k = \frac{1}{d^+} \left( X_0^{(0)} + \sum_{i=1}^{d} X_i^{(0)} \right),$$

where the initial features of the focal node received the index $0$. As the initial features are all normally distributed and independent, their sum is again normally distributed, as previously discussed. Using the same arguments as in Theorem 1, we obtain $\mathbb{P}(X^{(1)} \leq \theta \mid C = c_1, D = d, K = k) = \Phi\left(\frac{\theta d^+ - k\mu}{\sqrt{d^+}}\right)$ and $\mathbb{P}(X^{(1)} > \theta \mid C = c_2, D = d, K = k) = \Phi\left(\frac{\theta d^+ - (d^+ - k)\mu}{\sqrt{d^+}}\right)$. Plugging these results into the above derivation completes the proof. $\qquad\square$

**Theorem** (Theorem 4 in main paper). *After one round of mean aggregation and deleting edges between nodes that share not the same predicted label, the expected SoLAR accuracy of mean aggregation with respect to the rewired graph becomes*

$$\mathbb{E}(\mathcal{A}) = \mathbb{P}\left( X_0^{(0)} + \sum_{i=1}^{d} S_i X_i^{(0)} \leq \left( \sum_{i=1}^{d} S_i + 1 \right) \frac{\mu}{2} \right),$$

*where $S_i \in \{0,1\}$ indicates whether the respectice is pruned ($S_i = 0$) or not ($S_i = 1$). The messages $Z_i = S_i X_i^{(0)}$ sent by neighbors are only independent given the initial node feature $X_0^{(0)}$ and neighbor features $X_j^{(0)}$. We have $S_i = 1$ if $X_0^{(0)} + \sum_{i=1}^{d} X_i^{(0)} \leq \frac{\mu}{2} d^+$ and $X_i^{(1)} = (X_0^{(0)} + X_i^{(0)} + Y_i) \leq d^+ \mu/2$ or if $X_0^{(0)} + \sum_{i=1}^{d} X_i^{(0)} > \frac{\mu}{2} d^+$ and $X_i^{(1)} = (X_0^{(0)} + X_i^{(0)} + Y_i) > d^+ \mu/2$, where $Y_i \sim 0.5\mathcal{N}(\mu k, d-1) + 0.5\mathcal{N}(\mu(d-k+1), d-1)$. $Z_i = S_i = 0$ otherwise.*

*Proof.* As the setting is completely symmetric, nodes of class $c_1$ have exactly the same probability to be correctly classified as nodes of class $c_2$. Furthermore, the shape of their feature distributions after one round of mean aggregation (with respect to the rewired graph) is also identical and their means maintain the same distance to $\mu/2$. In consequence, the Bayes optimal decision threshold remains $\theta = \mu/2$ and, without loss of generality, we can focus on the correct classification probability of a class $c_1$ node, which is given by the stated formula for the average accuracy, which is equivalent to $\mathbb{P}(X^{(1)} \leq \theta)$, where $\theta = \mu/2$.

In comparison to the structure before SoLAR pruning, the distribution of the messages received by the original neighbors has changed from $X_i^{(0)}$ to $Z_i = S_i X_i^{(0)}$ with binary $S_i \in \{0,1\}$ indicating whether the edge to the respective neighbor has been pruned ($S_i = 0$) or still remains intact ($S_i = 1$). Accordingly, the degree of the node has become a random variable $D_s = \sum_{i=1}^{d} S_i$. The main challenge is that the pruning decision and thus the $S_i$ depend on all the initial features of the neighbors so that the messages $Z_i = S_i X_i^{(0)}$ become dependent random variables, whose distribution we cannot simply compute with convolutions.

The theorem thus states the conditions when $S_i = 1$ and thus the edge stays intact because the focal node with index $0$ has received the same label as its neighbour $i$. This is the case in two scenarios, either both the node and the neighbor receive both class $c_1$ or both class $c_2$. These scenarios correspond to the stated conditions.

Only the distribution of the random variable $Y_i$ in the statement is left to derive. $Y_i$ accounts for the messages that the neighbor has received from its own neighbors (in addition to the message from the focal node $X_0^{(0)}$ and its own initial feature $X_i^{(0)}$) before SoLAR rewiring. We only have to consider the neighbour's state before SoLAR rewiring, because the label before SoLAR was used to decide whether an edge is pruned or not. The remaining messages are independently normally distributed. If the neighbor $i$ of the focal node has true label $c_1$ with probability $p = 0.5$, then $k$ of its neighbors have label $c_2$. One of its neighbors, the focal node, has class $c_1$. Thus, still $k$ of the other $d - 1$ neighbors have class $c_2$ with initial features with mean $\mu$. The second normal distribution in the mixture of the distribution of $Y_i$ corresponds to the case where $i$ has true label $c_2$ so that $d - k$ of its neighbors have true label $c_2$. The focal node (one of its neighbors) has true class $c_1$. Thus $d - k$ neighbors and the initial features of $i$ are all distributed according to normal distributions with mean $\mu$. $\qquad\square$

## B  TYPES OF EDGE IMPROVEMENTS FROM ONE-STEP TO AFTER-PRUNING

Below we detail statistics for the proportion of edges pruned and not pruned for the 4 real-world graphs (Cora, Citeseer, Squirrel, Chameleon) which compare one step of mean aggregation with the SoLAR-like pruning based on the one-step predictions. The most important trends of this data are discussed in subsection 3.3. Within the pruned/not pruned sets of edges, we count the amount of them such that they connect nodes originally well predicted, wrongly predicted, or one of each, after one step of mean aggregation. Next, we compare these predictions to the predictions made after the pruning process described in subsection 3.2. Concretely, we subdivide the previous edge sets depending on whether we flipped each nodes' predictions or if they remained the same. This uncovers which kind of nodes we are able to correctly classify after pruning, and which are misclassified after the process. We report each value as a proportion of edges with respect to the previous category in a tree-like structure. We provide a list to account for all values of $\mu$ considered, and next to it the mean of this list. In subsection 3.3 we only describe the relevant scenarios according to these averages, but all trends persist in general.

```
Cora
m=[0.25, 0.5, 1.0, 1.5]
|-- Pruned edges: [0.32, 0.27, 0.17, 0.09]: 0.21
|   |-- Originally both right: [0.02, 0.03, 0.07, 0.16]: 0.07
|   |   |-- Maintained both right: [0.82, 0.87, 0.94, 0.98]: 0.9
|   |   |-- One corrupted: [0.15, 0.11, 0.05, 0.02]: 0.08
|   |   +-- Both corrupted: [0.03, 0.02, 0.01, 0.0]: 0.02
|   |-- Originally both wrong: [0.01, 0.01, 0.01, 0.01]: 0.01
|   |   |-- Improved both right: [0.11, 0.17, 0.27, 0.28]: 0.21
|   |   |-- One improved: [0.27, 0.3, 0.32, 0.32]: 0.3
|   |   +-- None improved: [0.62, 0.54, 0.42, 0.4]: 0.5
|   +-- One right, one wrong: [0.97, 0.96, 0.92, 0.84]: 0.92
|       |-- Improved the wrong: [0.16, 0.19, 0.25, 0.28]: 0.22
|       |-- Maintained: [0.71, 0.69, 0.67, 0.66]: 0.68
|       |-- Opposite: [0.06, 0.06, 0.05, 0.05]: 0.05
|       +-- Corrupted the right: [0.07, 0.05, 0.03, 0.02]: 0.04
+-- Not pruned edges: [0.68, 0.73, 0.83, 0.91]: 0.79
    |-- Both right: [0.68, 0.81, 0.93, 0.97]: 0.85
    |   |-- Maintained both right: [0.95, 0.97, 0.99, 1.0]: 0.98
    |   |-- One corrupted: [0.04, 0.03, 0.01, 0.0]: 0.02
    |   +-- Both corrupted: [0.01, 0.0, 0.0, 0.0]: 0.0
    |-- Both wrong: [0.29, 0.17, 0.05, 0.01]: 0.13
    |   |-- Improved both right: [0.02, 0.03, 0.05, 0.06]: 0.04
    |   |-- One improved: [0.07, 0.07, 0.07, 0.05]: 0.06
    |   +-- None improved: [0.91, 0.89, 0.88, 0.89]: 0.89
    +-- One right, one wrong: [0.02, 0.02, 0.02, 0.01]: 0.02
        |-- Improved the wrong: [0.04, 0.05, 0.05, 0.04]: 0.05
        |-- Maintained: [0.93, 0.93, 0.94, 0.95]: 0.94
        |-- Opposite: [0.01, 0.01, 0.0, 0.0]: 0.0
```

```
            +-- Corrupted the right: [0.02, 0.01, 0.01, 0.0]: 0.01
Citeseer
m=[0.25, 0.5, 1.0, 1.5]
|-- Pruned edges: [0.32, 0.26, 0.15, 0.07]: 0.2
|   |-- Originally both right: [0.01, 0.02, 0.03, 0.06]: 0.03
|   |   |-- Maintained both right: [0.72, 0.79, 0.88, 0.94]: 0.83
|   |   |-- One corrupted: [0.21, 0.16, 0.11, 0.06]: 0.14
|   |   +-- Both corrupted: [0.07, 0.05, 0.02, 0.01]: 0.04
|   |-- Originally both wrong: [0.01, 0.01, 0.02, 0.03]: 0.02
|   |   |-- Improved both right: [0.13, 0.16, 0.2, 0.22]: 0.18
|   |   |-- One improved: [0.26, 0.28, 0.33, 0.41]: 0.32
|   |   +-- None improved: [0.61, 0.55, 0.47, 0.37]: 0.5
|   +-- One right, one wrong: [0.98, 0.97, 0.95, 0.91]: 0.95
|       |-- Improved the wrong: [0.16, 0.2, 0.26, 0.29]: 0.23
|       |-- Maintained: [0.71, 0.69, 0.66, 0.65]: 0.68
|       |-- Opposite: [0.06, 0.06, 0.05, 0.05]: 0.05
|       +-- Corrupted the right: [0.07, 0.05, 0.02, 0.01]: 0.04
+-- Not pruned edges: [0.68, 0.74, 0.85, 0.93]: 0.8
    |-- Both right: [0.69, 0.82, 0.93, 0.97]: 0.85
    |   |-- Maintained both right: [0.96, 0.97, 0.99, 1.0]: 0.98
    |   |-- One corrupted: [0.04, 0.03, 0.01, 0.0]: 0.02
    |   +-- Both corrupted: [0.01, 0.0, 0.0, 0.0]: 0.0
    |-- Both wrong: [0.29, 0.16, 0.05, 0.01]: 0.13
    |   |-- Improved both right: [0.02, 0.03, 0.04, 0.05]: 0.04
    |   |-- One improved: [0.07, 0.07, 0.05, 0.03]: 0.06
    |   +-- None improved: [0.92, 0.91, 0.91, 0.92]: 0.92
    +-- One right, one wrong: [0.03, 0.02, 0.02, 0.02]: 0.02
        |-- Improved the wrong: [0.02, 0.02, 0.02, 0.02]: 0.02
        |-- Maintained: [0.96, 0.96, 0.97, 0.98]: 0.97
        |-- Opposite: [0.01, 0.0, 0.0, 0.0]: 0.0
        +-- Corrupted the right: [0.01, 0.01, 0.01, 0.0]: 0.01
Squirrel
m=[0.25, 0.5, 1.0, 1.5]
|-- Pruned edges: [0.21, 0.21, 0.2, 0.17]: 0.2
|   |-- Originally both right: [0.24, 0.24, 0.23, 0.25]: 0.24
|   |   |-- Maintained both right: [0.7, 0.73, 0.8, 0.86]: 0.77
|   |   |-- One corrupted: [0.21, 0.19, 0.15, 0.11]: 0.16
|   |   +-- Both corrupted: [0.1, 0.08, 0.05, 0.03]: 0.06
|   |-- Originally both wrong: [0.25, 0.26, 0.26, 0.26]: 0.26
|   |   |-- Improved both right: [0.12, 0.12, 0.13, 0.12]: 0.12
|   |   |-- One improved: [0.25, 0.27, 0.28, 0.3]: 0.28
|   |   +-- None improved: [0.63, 0.61, 0.58, 0.58]: 0.6
|   +-- One right, one wrong: [0.51, 0.51, 0.5, 0.49]: 0.5
|       |-- Improved the wrong: [0.12, 0.12, 0.12, 0.11]: 0.12
|       |-- Maintained: [0.66, 0.66, 0.68, 0.72]: 0.68
|       |-- Opposite: [0.12, 0.11, 0.1, 0.07]: 0.1
|       +-- Corrupted the right: [0.11, 0.11, 0.1, 0.09]: 0.1
+-- Not pruned edges: [0.79, 0.79, 0.8, 0.83]: 0.8
    |-- Both right: [0.3, 0.32, 0.37, 0.4]: 0.35
    |   |-- Maintained both right: [0.92, 0.93, 0.95, 0.96]: 0.94
    |   |-- One corrupted: [0.04, 0.04, 0.03, 0.02]: 0.03
    |   +-- Both corrupted: [0.04, 0.03, 0.03, 0.02]: 0.03
    |-- Both wrong: [0.23, 0.2, 0.16, 0.13]: 0.18
    |   |-- Improved both right: [0.05, 0.05, 0.05, 0.04]: 0.05
    |   |-- One improved: [0.06, 0.06, 0.07, 0.05]: 0.06
    |   +-- None improved: [0.9, 0.89, 0.88, 0.91]: 0.9
    +-- One right, one wrong: [0.48, 0.48, 0.48, 0.47]: 0.48
        |-- Improved the wrong: [0.03, 0.02, 0.02, 0.02]: 0.02
        |-- Maintained: [0.91, 0.91, 0.92, 0.94]: 0.92
```

```
        |-- Opposite: [0.04, 0.04, 0.03, 0.02]: 0.03
        +-- Corrupted the right: [0.03, 0.03, 0.02, 0.02]: 0.02
Chameleon
m=[0.25, 0.5, 1.0, 1.5]
|-- Pruned edges: [0.16, 0.14, 0.11, 0.09]: 0.12
|   |-- Originally both right: [0.16, 0.17, 0.23, 0.28]: 0.21
|   |   |-- Maintained both right: [0.73, 0.79, 0.87, 0.93]: 0.83
|   |   |-- One corrupted: [0.17, 0.14, 0.09, 0.05]: 0.11
|   |   +-- Both corrupted: [0.1, 0.08, 0.04, 0.02]: 0.06
|   |-- Originally both wrong: [0.16, 0.17, 0.17, 0.18]: 0.17
|   |   |-- Improved both right: [0.16, 0.2, 0.24, 0.28]: 0.22
|   |   |-- One improved: [0.23, 0.24, 0.29, 0.33]: 0.27
|   |   +-- None improved: [0.61, 0.56, 0.47, 0.4]: 0.51
|   +-- One right, one wrong: [0.68, 0.66, 0.61, 0.54]: 0.62
|       |-- Improved the wrong: [0.12, 0.14, 0.16, 0.14]: 0.14
|       |-- Maintained: [0.66, 0.66, 0.68, 0.72]: 0.68
|       |-- Opposite: [0.12, 0.13, 0.09, 0.08]: 0.1
|       +-- Corrupted the right: [0.09, 0.08, 0.06, 0.06]: 0.07
+-- Not pruned edges: [0.84, 0.86, 0.89, 0.91]: 0.88
    |-- Both right: [0.41, 0.47, 0.54, 0.57]: 0.5
    |   |-- Maintained both right: [0.95, 0.96, 0.98, 0.99]: 0.97
    |   |-- One corrupted: [0.02, 0.02, 0.01, 0.01]: 0.02
    |   +-- Both corrupted: [0.02, 0.02, 0.01, 0.0]: 0.01
    |-- Both wrong: [0.24, 0.19, 0.12, 0.1]: 0.16
    |   |-- Improved both right: [0.05, 0.06, 0.06, 0.07]: 0.06
    |   |-- One improved: [0.03, 0.03, 0.02, 0.01]: 0.02
    |   +-- None improved: [0.92, 0.91, 0.92, 0.92]: 0.92
    +-- One right, one wrong: [0.34, 0.34, 0.33, 0.33]: 0.34
        |-- Improved the wrong: [0.01, 0.01, 0.01, 0.01]: 0.01
        |-- Maintained: [0.94, 0.95, 0.96, 0.96]: 0.95
        |-- Opposite: [0.03, 0.03, 0.03, 0.03]: 0.03
        +-- Corrupted the right: [0.01, 0.01, 0.01, 0.01]: 0.01
```

## C  EFFECT ON HOMOPHILY

Graph neural networks provably perform better on homophilic graphs and some *good-heterophilic* graphs (Ma et al., 2022). We investigate the effect our one-shot rewiring strategy (GCN☼GCN) has on Edge label informativeness (ELI) and adjusted homophily score proposed in (Platonov et al., 2023a) and report the Normalized Mutual Information between the node ground truth labels and community membership labels after performing modularity maximization (Clauset et al., 2004) on the rewired graph in Figure 5. Evidently, our rewiring strategy improves the homophily score, as well as the edge label informativeness (denoted by ELI), which is also found to have high correlation to GNN performance (Platonov et al., 2023a). We also better align the node ground truth labels to community labels, as we delete inter-community edges (denoted by NMI).

We also visualize a T-SNE plot in Figure 6 of the node embeddings after training on the original graph and the rewired graph (GCN☼GCN) on Cora and Squirrel datasets. From the figure, we can see that the classes are more separable in the embedding space on the rewired graph, the class separability is more evident in a homophilic graph like Cora (Figure 6(b)) than in a heterophilic graph like Squirrel (6(d)), highlighting the fact that GNNs are usually more useful in homophilic settings and if the surrogate model gives noisy labels for rewiring, the performance on the downstream is also affected.

## D  ADDITIONAL RESULTS

In Table 5 we compare our results with an additional baseline (Bi et al., 2024) (DHGR), which uses a feature similarity based rewiring for heterophilic graphs. As there is no code available to reproduce the results, we take the results reported from the paper. We also report results with SGC (Wu et al., 2019), which is a simplified version of the GCN (Kipf & Welling, 2017) with weight

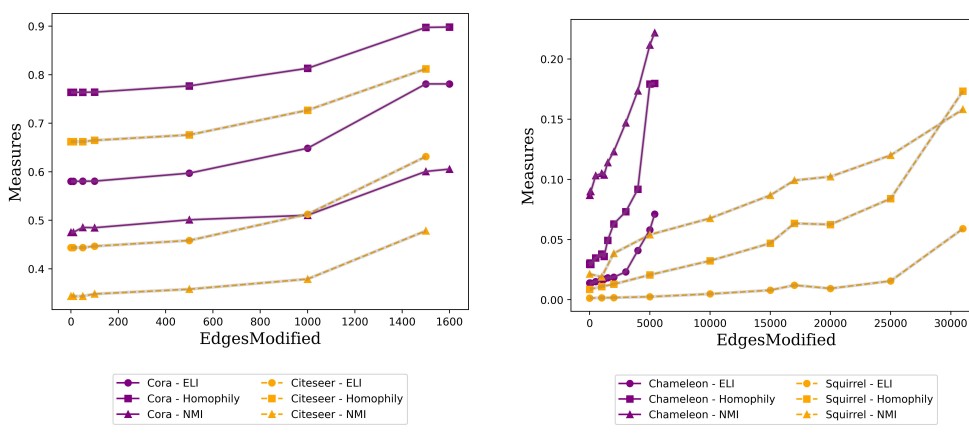

(a) ELI, Homophily, NMI for Cora and Citeseer with GCN✿GCN.

(b) ELI, Homophily, NMI for Chameleon and Squirrel with GCN✿GCN.

Figure 5: The effect of one-shot rewiring on ELI, homophily and NMI on Cora, Citeseer, Chameleon and Squirrel datasets.

matrices collapsed and non-linearities removed in Tables 6 and 7. In Table 8 we give results for simultaneous additions and deletions.

Table 5: Node classification on heterophilic graphs using one-shot rewiring.

| Method | Cornell | Texas | Wisconsin | Chameleon | Squirrel | Actor |
|---|---|---|---|---|---|---|
| GCN | 68.31±8.13 | 73.47±10.13 | 66.14±9.23 | 54.64±6.94 | 43.25±6.32 | 28.26±3.22 |
| GATv2 | 86.84±9.78 | 89.01±10.43 | 87.56±9.20 | 61.79±10.20 | 45.71±5.12 | 29.41±2.98 |
| GCN+FoSR | 71.64±9.80 | 73.93±10.23 | 65.85±7.73 | 54.40±6.58 | 42.80±6.40 | 28.66±3.21 |
| GATv2+FoSR | 76.12±6.51 | 78.15±7.81 | 74.08±9.01 | 46.48 ± 4.97 | 47.40±7.17 | 27.45±3.61 |
| GCN+ProxyAddMax | 67.57±1.71 | 81.08±1.75 | 70.00±1.61 | 56.74±0.90 | 33.26±0.39 | 27.57±0.22 |
| GCN+ProxyDelMax | 62.16±1.83 | 72.97±1.70 | 76.00±1.56 | 56.74±0.95 | 32.58±0.43 | 27.96±0.21 |
| GCN+DHGR | 67.38±5.33 | 81.78±0.89 | 76.47±3.62 | **70.83±2.03** | **67.15±1.43** | **36.29±0.12** |
| GATv2+DHGR | 70.09±6.77 | 83.78±3.37 | 73.20±4.89 | **72.11±2.87** | 62.37±1.78 | **34.71±0.48** |
| GCN✿GCN+Delete | 68.35±8.54 | 74.12±9.89 | 67.85±7.14 | 57.19 ± 6.45 | 44.50±6.29 | 29.25±3.50 |
| GCN✿GCN+Add | 69.42±8.93 | 74.20±10.26 | 68.51±7.20 | 56.43 ± 6.16 | 44.04±6.34 | 28.16±3.22 |
| GATv2✿GATv2+Delete | **87.40±9.89** | **90.14±10.64** | **88.32±9.08** | 68.89±11.50 | 49.10±5.59 | 30.31±4.29 |
| GATv2✿GATv2+Add | 87.12±9.59 | 87.97±10.95 | 87.76±9.57 | 66.35±11.18 | 46.44±6.00 | 29.46±4.67 |

Table 6: Node classification results on homophilic graphs with SGC.

| Method | Cora | Citeseer | Pubmed | CS | Physics | Photo |
|---|---|---|---|---|---|---|
| SGC | 88.78±0.48 | 80.51±0.59 | 82.47±0.41 | 93.39±0.18 | 95.21 ± 0.06 | 86.48±1.00 |
| GCN | 87.94±3.35 | 79.38±3.48 | 81.99±1.42 | 92.44±0.67 | 94.49 ± 0.04 | **92.89±1.23** |
| GCN✿SGCDelete | 88.10±0.48 | 80.14±0.64 | 82.12±0.32 | 93.68±0.13 | **94.97±0.03** | 89.93±0.83 |
| GCN✿SGCAdd | 89.02±0.48 | 79.14±0.72 | 82.06±0.37 | 93.43±0.18 | OOM | 87.15±0.98 |
| GATv2✿SGCDelete | **89.55±0.56** | **82.28±0.89** | **82.55±0.36** | **93.77±0.22** | 94.48±0.07 | 89.96±0.89 |
| GATv2✿SGCAdd | 89.16±0.50 | 80.85±0.83 | 81.96±0.38 | 93.44±0.18 | OOM | 87.26±0.98 |

# E    TRAINING DETAILS

We use PyTorch-Geometric (Fey & Lenssen, 2019) and DGL library (Wang et al., 2019) for all our experiments. We use a 2-layered GCN (Kipf & Welling, 2017) and GATv2 (Brody et al., 2022) with $\{8, 16\}$ attention heads. For datasets Cora, Citeseer, Pubmed, Cornell, Texas, Wisconsin, Chameleon, Squirrel, Actor, CS, Physics and Photo the final test accuracy is reported averaged over 100 splits, run for 100 epochs. We use the split mechanism introduced in (Shchur et al., 2019).

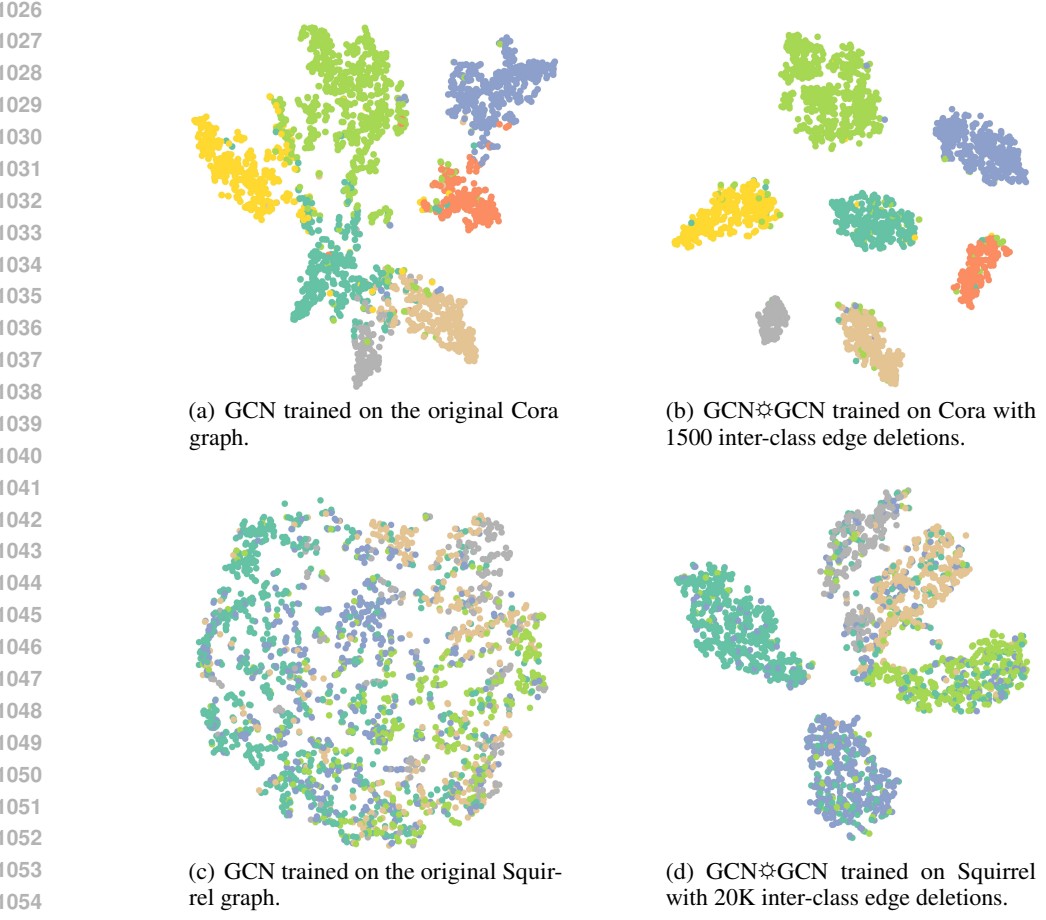

(a) GCN trained on the original Cora graph.

(b) GCN☼GCN trained on Cora with 1500 inter-class edge deletions.

(c) GCN trained on the original Squirrel graph.

(d) GCN☼GCN trained on Squirrel with 20K inter-class edge deletions.

Figure 6: We plot T-SNE for Cora and Squirrel datasets after training a GCN on the original graph and the rewired graph.

Table 7: Node classification results on heterophilic graphs with SGC.

| Method | Cornell | Texas | Wisconsin | Chameleon | Squirrel | Actor |
|---|---|---|---|---|---|---|
| SGC | 65.14±1.70 | 73.70±1.70 | 66.04±1.40 | 55.26±1.12 | 45.16±1.12 | 29.23 ±0.55 |
| GCN | 68.31±8.13 | 73.47±10.13 | 66.14±9.23 | 54.64±6.94 | 43.25±6.32 | 28.26±3.22 |
| GCN☼SGCDelete | 67.89±1.75 | 74.89±2.04 | 69.37±1.19 | 57.79±1.29 | 45.85±1.35 | 28.32±0.57 |
| GCN☼SGCAdd | 68.39±1.89 | 74.63±1.95 | 67.53±1.38 | 53.87±1.26 | 43.08±1.25 | 26.85±0.52 |
| GATv2☼SGCDelete | 75.86±1.86 | 83.13±2.13 | 74.04±1.40 | **66.82±2.11** | **47.71±1.35** | **30.32±0.83** |
| GATv2☼SGCAdd | **83.73±2.16** | **86.40±2.28** | **81.09±1.83** | 64.20±2.07 | 45.45±1.22 | 27.01±0.60 |

The weight decay and dropout are set to 0. The hidden dimension sizes we experimented with are {32,128,512} and learning rate {0.01,0.001}. The heterophilic graphs (Cornell, Texas, Wisconsin, Chameleon, Squirrel and Actor) are taken from (Platonov et al., 2023b). For experiments on Roman-empire and Amazon-ratings, we use the code base provided by (Platonov et al., 2023b), where the datasets are split into 50/25/25 for train/test/validation respectively. The accuracy is averaged over 10 runs run for 1000 epochs. We use a 5-layered GCN and GATv2 for these experiments, which are further augmented with skip connections, layernorm (Ba et al., 2016) and batchnorm Ioffe & Szegedy (2015) to facilitate training them better. For the Penn94 dataset introduced in (Lim et al., 2021), we use hidden dimension size of 32, learning rate set to $0.01$, weight decay $1e-3$ and also batchnorm. All the experiments were done on 2 V100 GPUs. The hyperparameters used for our experiments are provided in the tables below. The runtime is provided in seconds for one-shot rewiring. The statistics for the datasets used are given in Table 9. Our code is available here: https://anonymous.4open.science/r/SoLAR4356/README.md.

Table 8: Experiments for simultaneous additions and deletions.

| Method | Cora | Citeseer | Pubmed | CS | Photo | Physics | Cornell | Texas | Wisconsin | Chameleon | Squirrel | Actor |
|---|---|---|---|---|---|---|---|---|---|---|---|---|
| GCN | 87.94 ±3.35 | 79.38 ±3.48 | 81.99 ±1.42 | 92.44 ±0.67 | 92.89 ±1.23 | 93.64 ±0.16 | 68.31 ±8.13 | 73.47 ±10.13 | 66.14 ±9.23 | 54.64 ±6.94 | 43.25 ±6.32 | 28.26 ±3.22 |
| GATv2 | 89.13 ±3.13 | 81.92 ±4.81 | 81.83 ±1.04 | 91.90 ±1.59 | 91.22 ±2.18 | 94.07 ±0.44 | 86.84 ±9.78 | 89.01 ±10.43 | **87.56** ±**9.20** | 61.79 ±10.20 | 45.71 ±5.12 | 29.41 ±2.98 |
| GCN☼GCN | 89.10 ±0.51 | 78.81 ±0.80 | 81.73 ±0.26 | 93.30 ±0.12 | 93.28 ±0.18 | 94.26 ±0.04 | 68.11 ±2.04 | 75.45 ±1.89 | 68.01 ±1.52 | 55.06 ±1.31 | 43.11 ±1.29 | 27.33 ±0.57 |
| GATv2☼GATv2 | 89.16 ±0.60 | 81.13 ±0.95 | 81.13 ±0.27 | 93.42 ±0.21 | 93.89 ±0.31 | 94.15 ±0.09 | **86.72** ±**2.02** | 88.96 ±2.05 | 87.17 ±1.89 | **68.29** ±**2.32** | 47.63 ±1.23 | 29.70 ±0.95 |
| GCN☼GATv2 | **90.10** ±**0.57** | **81.43** ±**0.81** | **82.99** ±**0.27** | **93.94** ±**0.15** | 86.62 ±2.19 | **94.62** ±**0.03** | 79.93 ±2.08 | 85.05 ±2.24 | 84.05 ±1.78 | 56.15 ±1.29 | 48.26 ±1.18 | **29.92** ±**0.72** |
| GATv2☼GCN | 89.25 ±0.56 | 81.09 ±0.96 | 81.63 ±0.26 | 93.59 ±0.20 | **93.48** ±**0.48** | 94.49 ±0.06 | 85.55 ±2.13 | **89.16** ±**2.16** | 85.45 ±1.94 | 67.75 ±2.27 | **48.36** ±**1.28** | 29.78 ±0.95 |

Table 9: Statistics of the graphs used. We use the largest connected component for all our experiments.

| Dataset | #Nodes | #Edges |
|---|---|---|
| Cora | 2,708 | 10,138 |
| Citeseer | 3,327 | 7,358 |
| Pubmed | 19,717 | 88,648 |
| Cornell | 183 | 277 |
| Texas | 183 | 279 |
| Wisconsin | 251 | 450 |
| Chameleon | 890 | 8,854 |
| Squirrel | 2,223 | 57,850 |
| Actor | 7,600 | 26,659 |
| CS | 18,333 | 1,63,788 |
| Physics | 34,493 | 4,95,924 |
| Photo | 7,650 | 2,38,162 |
| Roman-empire | 22,662 | 32,927 |
| Amazon-ratings | 24,492 | 93,050 |
| Penn94 | 41,554 | 13,62,229 |

Table 10: Hyperparameters for GCN☼GCN+Del

| Dataset | EdgesDeleted | LR | HiddenDimension | Runtime |
|---|---|---|---|---|
| Cora | 1500 | 0.01 | 32 | 71.43 |
| Citeseer | 1500 | 0.01 | 32 | 84.08 |
| Pubmed | 10000 | 0.01 | 32 | 90.79 |
| Cornell | 100 | 0.001 | 128 | 86.76 |
| Texas | 100 | 0.001 | 128 | 73.94 |
| Wisconsin | 100 | 0.001 | 128 | 77.23 |
| Chameleon | 5400 | 0.001 | 128 | 76.82 |
| Squirrel | 310000 | 0.001 | 128 | 78.70 |
| Actor | 16000 | 0.001 | 128 | 80.12 |
| CS | 22000 | 0.01 | 128 | 200.90 |
| Physics | 30000 | 0.01 | 128 | 412.47 |
| Photo | 35000 | 0.01 | 512 | 263.11 |

Table 11: Hyperparameters for GCN☼GCN+Add

| Dataset | EdgesAdded | LR | HiddenDimension | Runtime |
|---|---|---|---|---|
| Cora | 6929 | 0.01 | 32 | 89.43 |
| Citeseer | 7168 | 0.01 | 32 | 70.88 |
| Pubmed | 352 | 0.01 | 32 | 94.07 |
| Cornell | 55 | 0.001 | 128 | 88.76 |
| Texas | 54 | 0.001 | 128 | 74.33 |
| Wisconsin | 41 | 0.001 | 128 | 86.68 |
| Chameleon | 4088 | 0.001 | 128 | 70.35 |
| Squirrel | 12349 | 0.001 | 128 | 74.85 |
| Actor | 12215 | 0.001 | 128 | 78.38 |
| CS | 8680 | 0.01 | 32 | 129.36 |
| Physics | 45991 | 0.01 | 32 | 351.85 |
| Photo | 26846 | 0.01 | 32 | 108.79 |

Table 12: Hyperparameters for GATv2☼GATv2+Add

| Dataset | EdgesAdded | LR | HiddenDimension | Runtime |
|---|---|---|---|---|
| Cora | 9711 | 0.001 | 32 | 149.73 |
| Citeseer | 11996 | 0.001 | 32 | 192.35 |
| Pubmed | 17647 | 0.001 | 32 | 594.85 |
| Cornell | 37 | 0.001 | 32 | 134.70 |
| Texas | 55 | 0.001 | 32 | 123.80 |
| Wisconsin | 49 | 0.001 | 32 | 135.06 |
| Chameleon | 4167 | 0.001 | 32 | 105.65 |
| Squirrel | 20754 | 0.001 | 32 | 313.19 |
| Actor | 30251 | 0.001 | 32 | 388.99 |
| CS | 27592 | 0.001 | 32 | 2292.85 |
| Physics | 46700 | 0.001 | 32 | 1761.90 |
| Photo | 27713 | 0.01 | 32 | 456.02 |

Table 13: Hyperparameters for GATv2☼GATv2+Del

| Dataset | EdgesDeleted | LR | HiddenDimension | Runtime |
|---|---|---|---|---|
| Cora | 1700 | 0.001 | 32 | 105.27 |
| Citeseer | 1500 | 0.001 | 32 | 116.28 |
| Pubmed | 14126 | 0.001 | 32 | 395.73 |
| Cornell | 120 | 0.001 | 32 | 100.89 |
| Texas | 120 | 0.001 | 32 | 131.33 |
| Wisconsin | 120 | 0.001 | 32 | 131.69 |
| Chameleon | 6000 | 0.001 | 32 | 169.09 |
| Squirrel | 35000 | 0.001 | 32 | 212.64 |
| Actor | 30000 | 0.001 | 32 | 139.54 |
| CS | 30000 | 0.001 | 32 | 1579.53 |
| Physics | 30000 | 0.001 | 32 | 3766.81 |
| Photo | 40264 | 0.001 | 32 | 450.34 |

