# OpenReview forum: "SoLAR: Surrogate Label Aware GNN Rewiring"
_ICLR.cc/2025/Conference — ICLR 2025 Conference Withdrawn Submission_

### Official Review · Reviewer_9mfW · 2024-10-29

**Soundness:** 2
**Presentation:** 2
**Contribution:** 1
**Rating:** 3
**Confidence:** 3

**Summary:**

The paper introduces SoLAR (Surrogate Label Aware GNN Rewiring), a simple yet effective method for improving Graph Neural Networks (GNNs) by optimizing graph structure using predictions from a surrogate model. The core idea is straightforward - first train a GNN on the original graph, then use its predictions to rewire the graph by adding edges between nodes predicted to share the same label and removing edges between nodes with different predicted labels. Finally, train a new GNN on this rewired graph structure.
Through theoretical analysis and extensive experiments, the authors demonstrate that this rewiring approach can improve graph homophily and boost model performance across various benchmarks, especially in scenarios with noisy graphs and limited labeled data. The method can also be applied iteratively, with each new model serving as the surrogate for further refinement of the graph structure. While conceptually simple, SoLAR achieves strong performance on multiple established datasets.

**Strengths:**

1. The paper's greatest strength lies in its clarity and accessibility. The authors present their method in a clear, step-by-step manner in Section 2, with an intuitive pipeline (Figure 1) showing how surrogate model predictions guide graph rewiring.

2. The experimental evaluation is thorough and well-designed, particularly in Section 4. The authors conduct extensive experiments across different noise levels, label rates, and graph types (both homophilic and heterophilic), demonstrating robust performance improvements over baselines.

3. From an engineering perspective, the method offers practical value through its simplicity and effectiveness. The approach requires minimal modifications to existing GNN architectures.

**Weaknesses:**

1. While the paper presents a well-executed study with solid empirical results, there are some limitations worth noting. The overall pipeline shares considerable similarity with existing graph structure learning approaches. As detailed in recent surveys on graph structure learning [1], the idea of iteratively refining graph structure using model predictions has been explored in various forms [2,3], making the core contribution somewhat incremental.

2. The theoretical analysis, while providing useful insights about homophily, could benefit from deeper investigation. Some recent works [4,5] in the field, such as those analyzing linear separability in graph convolution with contextual stochastic block models, offer examples of more comprehensive theoretical analysis and a more general setting.

3. While the authors position graph rewiring as a pre-processing step to address over-smoothing and over-squashing issues in GNNs, there is limited discussion or analysis of how the proposed method actually tackles these fundamental challenges. The paper demonstrates improved performance on standard metrics but doesn't explicitly show how the rewiring process mitigates over-smoothing or alleviates the information bottleneck causing over-squashing.

[1] Zhu, Yanqiao, et al. "A survey on graph structure learning: Progress and opportunities." arXiv preprint arXiv:2103.03036 (2021).
[2] Dai, Enyan, et al. "Towards robust graph neural networks for noisy graphs with sparse labels." Proceedings of the Fifteenth ACM International Conference on Web Search and Data Mining. 2022.
[3] Stretcu, Otilia, et al. "Graph agreement models for semi-supervised learning." Advances in Neural Information Processing Systems 32 (2019).
[4] Baranwal, Aseem, Kimon Fountoulakis, and Aukosh Jagannath. "Graph convolution for semi-supervised classification: Improved linear separability and out-of-distribution generalization." arXiv preprint arXiv:2102.06966 (2021).
[5] Baranwal, Aseem, Kimon Fountoulakis, and Aukosh Jagannath. "Effects of graph convolutions in multi-layer networks." arXiv preprint arXiv:2204.09297 (2022).

**Questions:**

1. The paper mentions addressing over-smoothing and over-squashing through graph rewiring. Could you provide theoretical or empirical analysis showing how SoLAR specifically helps with these issues?

2. Have you conducted experiments measuring the over-smoothing effect (e.g., feature similarity across layers) before and after applying SoLAR?

3. What is the computational overhead of the iterative refinement process on large-scale graphs?

4. Have you considered comparing with more recent graph structure learning methods?

5. How does the method perform when the graph structure contains different types of noise beyond random edges?

---

### Official Review · Reviewer_Rp5z · 2024-10-31

**Soundness:** 3
**Presentation:** 3
**Contribution:** 3
**Rating:** 6
**Confidence:** 4

**Summary:**

This is a paper on "Graph Rewiring", i.e. on changing the topology of the graph to improve the downstream task.
The proposed method (SoLAR or "Surrogate Label Aware Rewiring") reuses the predicted node labels from a surrogate model for rewiring.
The strategy is to "Iterative SoLAR, which alternates between model training and graph rewiring cycles, yields further performance boosts".

Theoretically, this process can be understood as improving the homophily of the original graph. This does not mean that the graph should
be homophilic to work well, simply that an heterophilic graph is "homophiliated". This usually requires high-order knowledge as it is the
case.

After one round of mean aggregation and deleting edges between nodes that do not share the same predicted label, SoLAR shows that the expected accuracy relies on the pruned nodes.

The experiments are consistent and useful to understand the approach.

**Strengths:**

Easy to capture the main idea and nice formalism.

**Weaknesses:**

-The intuition that increasing homophily may be only useful for homophilic graph. Maybe move Fig.4 to previous sections to give an intuition of what is happening.

**Questions:**

The 60/20/20 split is too terse. How SoLAR does work for the 48/32/20?

---

### Official Review · Reviewer_sebW · 2024-11-04

**Soundness:** 1
**Presentation:** 3
**Contribution:** 1
**Rating:** 3
**Confidence:** 3

**Summary:**

The authors proposed a method, namely SoLAR to improve node classification tasks on graphs, by rewiring the graph in a way to better match the graph with the task. the reasoned there are two mechanisms here, one is denoising, the second is a KD-like process.

**Strengths:**

The authors proposed to rewire the graph as an approach to preprocess the graph (task-specific) to improve learning performance. The key idea is to further exploit the homophily on the graph, which is a critical mechanism in node classification. The paper is clearly written.

**Weaknesses:**

I have concerns about the novelty and soundness of the methods being proposed here.

The basic process of SoLAR is to do two-passes, the first-pass predicted labels are then used to add intra-class edges and/or delete inter-class edges to improve homophily. Then the second model learns from the rewired graph.

1. From another angel, the operation is a way to denoising: predict/impute (the first-pass predicted labels are only used on test/val because these labels are missing, and the authors use the ground truth label on the training set, ln176-177) the labels on the unknown nodes, then globally remove/add edges to improve homophily.

2. The analysis in the paper is circular as a lot of effects are expected if edges are added/removed to improve homophily. For example, the authors showed a variance reduction effect; intuitively, the process here will make the model more confident of its prediction by rewiring the graph systematically in a way in concordance with the labels predicted.

**Questions:**

1. The method can be viewed as an imputating/denoising method, the authors should consider adding a section on related work and compare.
2. The method has quite different effects on homophic and hetrophilic graphs (fig 4), is this consistent in other graphs? Do we have any explanation on this?
3. The author proposed the idea of knowledge-distillation in this process but only included conceptual description and discussion; can the authors give more analyses into what exactly the teacher and student are learning there.

---

### Official Review · Reviewer_npPq · 2024-11-04

**Soundness:** 2
**Presentation:** 3
**Contribution:** 2
**Rating:** 5
**Confidence:** 4

**Summary:**

The paper presents SoLAR (Surrogate Label Aware Rewiring), a method that improves graph neural network (GNN) performance by adjusting graph connections based on label predictions from a surrogate model. Instead of only changing the graph’s topology, SoLAR adds edges between nodes with similar predicted labels and removes edges between nodes with different labels, increasing homophily and enhancing model accuracy.  Experiments show that SoLAR improves performance on various datasets.

**Strengths:**

1. The concept of graph rewiring is interesting.
2. The proposed method demonstrates promising results across multiple datasets, including a real-world application, highlighting its practical effectiveness and versatility.
3. The method is supported by some theoretical analysis, which strengthens the validity of the approach.

**Weaknesses:**

1. The method appears to be primarily effective on homophilic graphs, which limits its applicability in domains with heterophilic or complex graph structures. A discussion on its performance across varying levels of homophily would strengthen the understanding of its generalizability.

2. The approach is computationally intensive, as it requires an additional surrogate GNN model, significantly increasing computational costs. The scalability of the proposed method is not thoroughly explored, particularly for large-scale graphs where such an approach might be impractical.

3. The novelty of the proposed method appears limited, as the idea of enhancing homophily by adding same-prediction edges or removing different-label edges has been explored in several previous works.

**Questions:**

same as in Weakness

---

### Official Review · Reviewer_P7bd · 2024-11-05

**Soundness:** 3
**Presentation:** 2
**Contribution:** 2
**Rating:** 5
**Confidence:** 3

**Summary:**

This paper presents a graph rewiring technique. The authors propose to train a GNN on a dataset, then to rewire the graph based on the GNN’s outputs to increase homophily. After modifying the graph structure, a second GNN is trained on the rewired graph. The process may be repeated, though the main results seem to rely on two GNNs.

**Strengths:**

1) The experimental results presented in the paper appear to advance the state of the art.
2) Some theoretical development is provided to partially support the rewiring technique.

**Weaknesses:**

Theoretical:
---
1) Graph rewiring and graph structure learning are well-established areas of research, and more related work should be referenced. For example, the proposed technique resembles [1], where the graph is rewired after each convolution layer based on the features and trained end-to-end.
2) The rewiring technique is not explicitly described. It’s important to include details on how edges are added or removed, along with the relevant hyperparameters and their tuning process, in the main text.
3) In Table 9, the authors note that only the largest connected components are used for each dataset. Clarification is needed on why portions of the dataset were discarded, as this could greatly affect performance outcomes. This fact should also be highlighted in the main text.

Experimental:
---
4) The results are not reproducible with the provided code. After fixing minor errors, I attempted to replicate the results using the code in the appendix, but I could not achieve the reported outcomes. For instance, using `GATv2+GATv2+Delete` (`n_layers=2` each, `hidden_dim=32`, `lr=0.001`, `max_iters_delete=1700`), I obtained a final test accuracy of $44.41 ± 2.32 %$, compared to $90.06 ± 3.31 %$ reported in Table 1. While I do _not_ suspect unethical practices, the code quality requires improvement for reliable reproduction.
5) You state that predictions are used “only on the test and validation sets.” However, in the code (`methods.py` line 393) it appears that the true validation labels are also used during the rewiring process.

Minor:
---
The overall presentation of the paper could be improved. For instance, the parameters $(d, k)$ in Figure 2 are not defined. On page 5, "SoLAR pruning" is introduced and reused without explanation.

References:
---
[1] Wang, Yue, et al. "Dynamic graph cnn for learning on point clouds." ACM Transactions on Graphics (tog) 38.5 (2019): 1-12.

**Questions:**

To what extent do the theoretical results support the use of SoLAR? In particular,

1) Based on the findings in Section 3.1, why is SoLAR preferable to a simple GNN, especially considering that the discussion in the 'Benefits of homophily' section could apply to a generic GNN as well?

2) Moreover, the mean-field assumption and absence of parameterization may limit the practical applicability of the theoretical results. Could you provide some insight into how applicable you consider these results to be in real-world scenarios?

---

### Note · Authors · 2024-11-29

**Comment:**

We are grateful to the reviewers for their insightful feedback, which has greatly helped us improve our work. While addressing the comments, we identified an issue in our experimental setup that affects some of the results. Unfortunately, updating the experimental tables requires more time than has been available within the current timeline. We deeply value the reviewers' suggestions and we plan to incorporate them as we revise the paper. However, we believe the updated manuscript will require another round of review to ensure a fair assessment.

**Withdrawal Confirmation:**

I have read and agree with the venue's withdrawal policy on behalf of myself and my co-authors.